# Equivariant and Denoising CNNs to Decouple Intensity and Spatial Features for Motion Tracking in Fetal Brain MRI

**Benjamin Billot**[1]                                                     BBILLOT@MIT.EDU
**Daniel Moyer**[2]                                      DANIEL.MOYER@VANDERBILT.EDU
**Neerav Karani**[1]                                          NKARANI@CSAIL.MIT.EDU
**Malte Hoffmann**[3]                                      MHOFFMANN@MGH.HARVARD.EDU
**Esra Abaci Turk**[4]                          ESRA.ABACITURK@CHILDRENS.HARVARD.EDU
**P. Ellen Grant**[4]                            ELLEN.GRANT@CHILDRENS.HARVARD.EDU
**Polina Golland**[1]                                          POLINA@CSAIL.MIT.EDU

[1] *Massachusetts Institute of Technology, Cambridge, MA, USA*

[2] *Vanderbilt University, Nashville, TN, USA*

[3] *Massachusetts General Hospital and Harvard Medical School, Boston, MA, USA*

[4] *Boston Children's Hospital and Harvard Medical School, Boston, MA, USA*

## Abstract

Equivariance in convolutional neural networks (CNN) has been a long-sought property, as it would ensure robustness to expected effects in the data. Convolutional filters are by nature translation-equivariant, and rotation-equivariant kernels were proposed recently. While these filters can be paired with learnable weights to form equivariant networks (E-CNN), we show here that such E-CNNs have a limited learning capacity, which makes them fragile against even slight changes in intensity distribution. This is a major issue in medical imaging where many noise sources can randomly corrupt the data, even for consecutive scans of the same subject. Here, we propose a hybrid architecture that successively decouples intensity and spatial features: we first remove irrelevant noise in the data with a denoising CNN, and then use an E-CNN to extract robust spatial features. We apply our method to motion tracking in fetal brain MRI, where it considerably outperforms CNNs and E-CNNs.

**Keywords:** Equivariant CNN, Denoising, Motion tracking, Fetal brain MRI

## 1. Introduction

Modern image processing almost exclusively relies on convolutional neural networks (CNNs), which build hierarchical representations of images by applying learned convolutional filters. A long-desired property is to make these kernels equivariant to spatial transforms expected to occur in the data. For example, convolutional filters are by nature equivariant to translations: the outputs shift accordingly with the inputs. However, constructing kernels that are equivariant to transforms other than translation is challenging. Thus, equivariance is usually *learned* with data augmentation, and possibly contrastive learning (Chen et al., 2020). If these strategies generally improve results, they do not explicitly ensure equivariance.

Research on equivariant CNN filters has mainly focused on rigid transforms (i.e., translations and rotations). While initial methods worked for discrete angles only (Winkels and Cohen, 2018; Bekkers et al., 2018), fully rigid-equivariant filters were introduced by Weiler et al. (2018). Since such filters are pre-computed and fixed, Moyer et al. (2021) recently proposed to pair them with learnable weights to form trainable equivariant CNNs (E-CNNs).

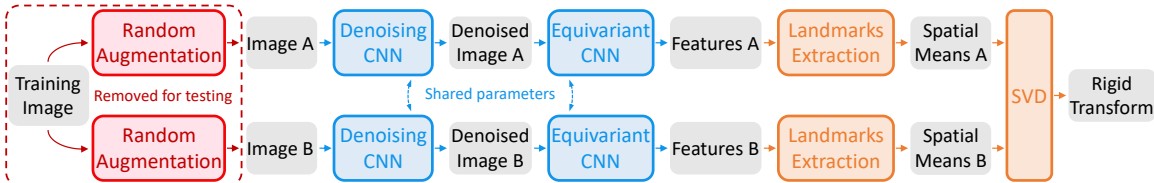

Figure 1: Overview of the proposed framework for registration-based 3D motion tracking.

These networks showed promising results when applied to motion tracking in fetal brain MRI (Moyer et al., 2021). However, they were only tested on simulated data that ignored intensity and noise variations across scans. Here, we evaluate E-CNNs on real data, showing that they have a limited learning capacity, which makes them sensitive to intensity changes. This is a major issue in medical imaging and especially in fetal MRI, where two consecutive scans may differ substantially due to scanner noise, motion artefacts, histogram shifting, etc.

Here, we present a new method to decouple intensity and spatial features for registration-based tracking. We first use a CNN to remove sources of noise in the data, and then obtain expressive spatial features with an E-CNN. While most disentangling strategies treat intensity and spatial features in parallel (Chartsias et al., 2019), we process them successively to fully leverage the potential of E-CNNs. We demonstrate our method in MRI fetal brain motion tracking, where it yields considerably better results than CNNs and E-CNNs.

## 2. Methods

**Augmentation:** A training step starts by randomly selecting a scan and independently augmenting it twice to simulate differences between scans at testing (Fig.1). We first randomly translate ($[-30, 30]$ voxels) and rotate ($[-\pi, \pi]$ rad) along all axes, and then add random noise, motion artefacts, bias field, and histogram shifting (Pérez-García et al., 2021).

**Denoiser:** We then employ a denoising CNN (D) to remove the previously injected noise from the augmented images. This step seeks to remove all anatomically irrelevant intensity features, such that the following E-CNN can extract robust spatial features. Here we use a UNet (Ronneberger et al., 2015) with 5 levels, each with 2 convolutional layers (32 kernels of size $3 \times 3 \times 3$), ReLU non-linearities, and batch normalisation (except for the last layer).

**E-CNN:** Denoised images are passed to an E-CNN for robust spatial feature prediction. We use rigid equivariant filters of size $5 \times 5 \times 5$ corresponding to discretised spherical harmonics of order 0, 1 and 2 (Weiler et al., 2018). Layers are formed by linearly combining the kernels, where the linear coefficients are the learnable parameters. Here, we employ 5 equivariant layers, each separated with equivariant ReLU non-linearities (Weiler et al., 2018).

**Rigid transform prediction:** We then compute the spatial means (i.e., centre of mass) of all the E-CNN output features. Finally, we derive a rigid transform between these two sets of landmarks (one for each scan) with singular value decomposition (SVD) (Horn, 1987).

**Training:** The denoiser D and the equivariant network are trained separately with the Adam optimiser and a learning rate of $10^{-5}$. The denoiser D is trained to remove the effect of intensity augmentation with an $L_2$ loss. The E-CNN is optimised using a geodesic loss (Salehi et al., 2018) between the ground truth and predicted transforms.

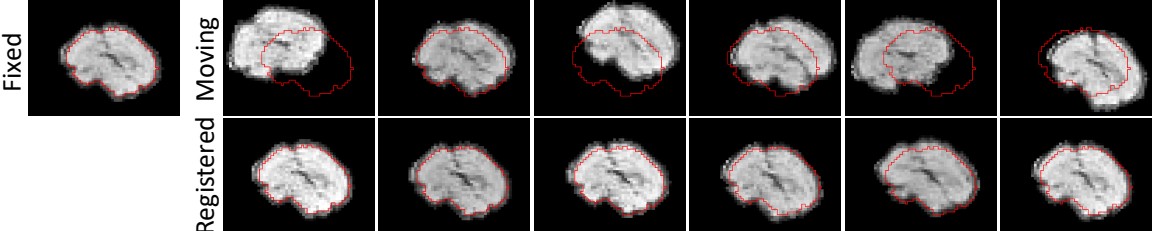

Figure 2: Original fetal brain MRI time-series (top) re-aligned by our method (bottom).

Table 1: Scores obtained for Exp. 1 (rotation, translation, Dice 1) and Exp. 2 (Dice 2).

| Methods | Angle error (°) | Shift error (vox.) | Dice 1 | Dice 2 |
|---|---|---|---|---|
| Conv. (de Vos et al., 2019) | 14.7 ± 4.2 | 5.2 ± 1.2 | 0.78 | 0.77 |
| E-CNN (Moyer et al., 2021) | 11.2 ± 3.4 | 4.1 ± 0.9 | 0.83 | 0.84 |
| Augm. + E-CNN | 10.9 ± 3.2 | 3.8 ± 1.2 | 0.84 | 0.84 |
| Augm. + D + E-CNN (Ours) | **5.1 ± 1.9** | **2.5 ± 0.9** | **0.90** | **0.89** |

## 3. Experiments and Results

**Data:** We use a fetal dataset of whole-uterus MRI time-series from 50 pregnant mothers. Scans are acquired on a 3T Skyra Siemens scanner using EPI sequences at 3mm isotropic resolution. Automatic brain masking is applied (Hoffmann et al., 2021) and dilated by 5 voxels for uncertainty. We split the time-series into 30/5/15 for training/validation/testing.

**Baselines:** We compare our method against a widely used CNN baseline for rigid registration ("Conv", de Vos et al. (2019)). We then perform ablations by successively removing the denoiser ("Augm+E-CNN") and the augmentation ("E-CNN", Moyer et al. (2021)).

**Results:** We first test all methods on simulated data obtained by augmenting the test scans as during training, such that we know the ground truth transforms (Exp. 1). While the pure E-CNN yields slightly better scores than traditional CNNs, its learning capacity remains limited, as using augmentation only leads to marginal improvements (Tab. 1). In comparison, employing a denoiser enables us to considerably outperform all methods.

These results are confirmed when testing on the real time-series (Exp. 2), where our method yields superior brain overlap between registered consecutive frames (Tab. 1, Fig. 2).

## 4. Discussion and Conclusion

Building on rigid-equivariant networks, we presented a new registration-based motion tracking strategy that leverages a denoising CNN to decouple intensity and spatial features. Our method substantially outperforms traditional CNNs and E-CNNs for motion tracking of fetal brain MRI, and has the potential to be deployed online to improve fetal acquisitions.

**Acknowledgements:** This research is supported by NIH NIBIB NAC P41EB015902, NIH NICHD R01HD100009, NIH NIBIB 5R01EB032708, NIH NICHD R00HD101553, and the Swiss National Science Foundation.

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
