# OpenReview forum: "Equivariant and Denoising CNNs to Decouple Intensity and Spatial Features for Motion Tracking in Fetal Brain MRI"
_MIDL.io/2023/Short_Paper_Track — MIDL 2023 Short paper track Poster_

### Official Review · Reviewer_RddC · 2023-04-21
**equivariant and denoting CNNs review**

**Rating:** 7
**Confidence:** 4

**Review:**

This short paper presents and extension of rotationally equivariant CNNs to extract spatial features only (ignoring non-anatomical intensity variations such as the bias field) by using a U-net in front. The U-net aims to remove the non-anatomical intensity combinations. Both components (UNet and ECNN) are previously existing architectures but this paper demonstrates their combined use to obtain improved results in fetal MRI registration. The improvement seems significant and the paper should have interest to research in this area.

---

### Official Review · Reviewer_MqnH · 2023-04-23
**Review E-CNNs for motion tracking in fetal brain MRI**

**Rating:** 6
**Confidence:** 5

**Review:**

This work showcases the limitations of equivariant convolutional neural networks (E-CNNs) when applied to real data, which according to the authors are sensitive to intensity changes.

Strengths: The motivation (E-CNN might be suboptimal in real data) is clear, which is supported by the experiments in Table 1. Integrating the proposed components has the potential to improve the performance of E-CNNs.

Weaknesses: It seems that the necessary changes to improve E-CNNs comes from pre-processing the data by either data augmentation or/and denoising images. Thus, I am not sure whether including these two steps should be considered as a new architecture. Furthermore, I found that the baseline D + E-CNN is missing in Table 1, which may give a better overview of the impact of this step. Last, it is unclear how existing methods leverage intensity and spatial features in parallel.

Despite the weaknesses, I think this work may generate interesting discussions in a poster session.